# Digestive Neurobiology in Autism: From Enteric and Central Nervous System Interactions to Shared Genetic Pathways

**DOI:** 10.3390/ijms26199580

**Published:** 2025-10-01

**Authors:** Raz Robas, Utkarsh Tripathi, Wote Amelo Rike, Omveer Sharma, Shani Stern

**Affiliations:** 1Department of Neurobiology, University of Haifa, Haifa 3498838, Israel; razrobas1@gmail.com (R.R.); wotepharma@yahoo.com (W.A.R.); os10@iitbbs.ac.in (O.S.); 2The Haifa Brain and Behavior Hub (HBBH), University of Haifa, Haifa 3498838, Israel

**Keywords:** autism spectrum disorder, enteric nervous system, gastrointestinal tract, digestive disorders, iPSCs

## Abstract

Social communication difficulties characterize autism spectrum disorders (ASD). Gastrointestinal (GI) symptoms are more common in ASD than in the general population. The identification of GI problems in individuals with ASD is challenging due to their altered pain perception and irregular behaviors. Importantly, GI symptoms and ASD can potentially aggravate each other. However, it is unclear if GI problems cause ASD symptoms or vice versa. A crosstalk between the digestive system, gut microbiota, and the central and enteric nervous systems (CNS and ENS, respectively) has been repeatedly reported. The ENS regulates the GI tract with the CNS and the autonomic nervous system (ANS), as well as independently through specific neural circuits. Several mechanisms contribute to GI problems in ASD, including genetic mutations that affect the ENS, dysregulation of the ANS, alterations in gut microbiota, unhealthy dietary preferences, and changes in metabolomic profiles. Furthermore, studies have shown molecular and cellular differences in the GI biopsy of children with and without ASD. These findings highlight the unique nature of GI issues in ASD, underscoring the importance of further investigating the changes that occur in the digestive system and ENS in ASD models.

## 1. Introduction

Autism spectrum disorder (ASD) is a neurodevelopmental condition characterized primarily by challenges in social interaction and communication, along with restricted and repetitive behaviors that can range from mild to severe [1]. It is a growing public health concern because ASD affects approximately 1:59 among children and 1:100 among adults [2]. In addition to the extensive symptoms and the various manifestations of ASD, the definitions of ASD have significantly changed, and new categories have been added over the years, including Asperger’s syndrome [3]. Studies also show that a variety of mechanisms are responsible for ASD. Despite years of research, the biological mechanisms underlying ASD remain only partially understood.

According to research findings about twins, some of the likelihood of developing ASD is due to ASD-related genetic variation [4]. However, there is no single genetic mutation that can explain more than 1–2% of ASD cases [5]. Studies have shown a wide range of predisposing ASD-associated mutations and polymorphisms [5,6,7,8,9,10,11,12]. Researchers poorly understand how these mutations lead to ASD symptoms, and multiple ASD-related genes are also associated with other neuropsychiatric disorders. For instance, in 2024, we found that approximately 75% of the ASD-associated genes in genome-wide association studies (GWAS) are also associated with schizophrenia [13]. Additionally, there are ASD-associated genes also related to attention deficit hyperactivity disorder, major depression, and intellectual disability.

Individuals with ASD have various comorbidities, including gastrointestinal (GI) symptoms. Studies indicate that individuals with ASD are significantly more likely than the general population to experience GI issues, such as food intolerances and sensitivities, nausea, vomiting, diarrhea, abdominal discomfort, flatulence, reflux, ulcers, inflammatory bowel diseases (IBD), and constipation [14,15]. Reports from recent years estimate that 46–84% of children with ASD experience GI symptoms, significantly higher than about 26–28% of children without ASD (the rate is not different between genders) [15,16,17]. Additionally, 20–25% of children with ASD are allergic to various types of food compared to 5–8% of children without ASD [18]. The high rates of GI problems in ASD patients may indicate a causative relationship. However, it is not clear if GI problems are a consequence of ASD pathology or if they contribute to ASD pathogenesis.

Children with ASD, as well as those with developmental delays, tend to experience more GI issues than other children. However, due to core ASD symptoms—such as difficulties in verbal and nonverbal communication and altered pain perception—assessing and interpreting subjective GI symptoms (such as pain and discomfort) in autistic children is particularly challenging [19]. Certain behaviors are thought to be expressions of GI problems in children with ASD, such as facial grimacing, teeth-gritting, and excessive chewing (of food or items). Vocal behaviors such as sobbing, screaming, or delayed echolalia may accompany them. Behaviors such as applying pressure to the stomach may also be associated with GI issues. These behavioral characteristics may not be specific to GI problems [20]. Additionally, they can coexist with general ASD behavioral deficits, such as self-injurious behaviors, repetitive or stereotypic movements, unusual posturing, and tapping/twitching [19]. As a result, physicians may inadequately treat GI disorders in this population, as they often attribute them to general ASD-related behaviors [21]. Moreover, individuals with ASD who experience GI symptoms tend to exhibit more severe ASD-related symptoms, including increased irritability, anxiety, and social withdrawal [22]. This review outlines current knowledge on the relationship between GI disorders and ASD, mainly in non-syndromic/idiopathic ASD. In contrast to other review articles that primarily scan the gut microbiota as a factor for ASD in general and GI disorders in ASD in part [23,24,25,26], this review considers a bidirectional system with many additional factors of GI disorders in ASD, such as the enteric nervous system (ENS) and genetic factors, and highlights the relationship between all of them. Genetic factors can lead to syndromic ASD. Still, some of these genes can also be expressed in idiopathic ASD. Moreover, we explain the immunological and genetic differences between GI disorders in ASD and GI disorders not in ASD. We will begin with a general overview of the gut–brain–microbiota axis. Later, we will describe separately the factors related to GI issues in ASD that involve the nervous system, the metabolic signaling pathways, and the immune system.

## 2. Methods

### 2.1. Data Collection

The articles were searched using the following terms (together and separately) in Google Scholar: ASD, ENS, gut–brain axis, gut microbiota, metabolomics, imaging, colonoscopy, IBD, limitations of maternal immune activation (MIA), limitations of valproic acid (VPA)-induced ASD (both are models of ASD), ASD and iPSCs (induced pluripotent stem cells). The search was not limited to specific years, but preference was given to articles between 2019 and 2025.

### 2.2. Pathway Analysis

ASD-associated genes were found in the Genome-Wide Association Studies (GWAS) catalog. We sought the genes expressed in the ENS, searching Genotype-Tissue Expression (GTEx) data (ASD-associated vs. not ASD-associated). Both lists of genes—ASD-associated genes and genes expressed in the ENS—were written in official gene symbols. The lists were cross-referenced to a new list of ASD-associated genes expressed in the ENS using a script written in the Python (version 3.13) programming language. Pathway analysis of the created list (Appendix A) of genes was performed using the databases: Gene Ontology: Biological Pathway (GO:BP), Cellular component (GO:CC), and Molecular Function (GO: MF), Protein Analysis Through Evolutionary Relationships: Biological Pathway (PARTNER:BP), Cellular Component (PANTHER:CC), and Molecular function (PANTHER: MF), Reactome, Kyoto Encyclopedia of Genes and Genomes (KEGG), and Motif.

## 3. The Gut-Microbiota-Brain Axis

Over the last years, preclinical and clinical studies have strongly supported the bidirectional crosstalk between the gut microbiota and the brain, occurring through parallel and interacting pathways (Figure 1) among individuals with and without ASD. The gut microbiota, the central nervous system (CNS), and the ENS communicate with each other through various mechanisms.

The vagus nerve is cranial nerve number 10, and it extends from the brainstem and innervates the viscera [26]. It comprises 90% afferent (sensory, ends in the muscular layer and the intestine’s mucosa) and 10% efferent (motor) neurons that transmit information directly between the gut and the brain. Vagal nerve stimulation—in the presence of certain bacterial strains and inflammatory contexts [27]—can attenuate systemic inflammatory responses via acetylcholine [28]. In addition, noradrenaline release from the sympathetic nervous system can indirectly influence the gut microbiota via altered goblet cell function [29]. The gut microbiota can secrete chemical stimuli, such as cytokines, nutrients, gut peptides, and hormones, or induce the release of these chemicals from enteroendocrine/GI immune cells [30]. The mucosal vagal afferent neurons and the CNS sense all these chemicals absorbed across the epithelial layer, and the gut microbiota can alter vagus nerve signaling in this manner.

Microbes in the gut also influence the development and function of immune cells, not just in the intestines, but also in the CNS and throughout the entire body.

For example, bacterial fermentation-derived short-chain fatty acids (SCFAs) cross the intestinal epithelial barrier and the BBB and regulate microglia’s homeostasis, maturation, and function [31]. The immune cells can modulate neural activities directly (by penetrating through the blood–brain barrier [BBB]) and indirectly (by connecting the vagus nerve and/or enteric nerves, which transmit information to the CNS) [32]. Some cytokines secreted by the gut microbiota can also cross the BBB, modulate inflammation in the CNS, and influence neural circuits [33]. For instance, interleukin-6 (IL-6) can cross the BBB through the placenta and fetal circulation. Therefore, elevated brain IL-6 during pregnancy (due to MIA) causes an imbalance between excitatory and inhibitory synapses and mediates autistic-like behaviors [33,34,35,36,37]. Microbial metabolites of tryptophan can also modulate CNS inflammation by activating astrocyte aryl hydrocarbon receptors [38].

Spore-forming microorganisms produce metabolites that cause colonic enterochromaffin cells to synthesize serotonin [39], and there are gut microbes that can synthesize neurotransmitters by themselves [40]. Some neurotransmitters can also act on the vagus nerve endings or enteric neurons. The gut microbiota is also crucial for the development and integrity of the BBB. For instance, germ-free mice exhibit increased BBB permeability, which can be corrected by colonizing gut microbiota [41]. The gut microbiota also mediates the postnatal development of the gut mucosal-epithelial layer and the lymphoid system, and protects from potential pathogens [42]. SCFAs, in addition to their association with microglia, serve as energy substrates for colonocytes, regulate colonic proliferation and differentiation, modulate the colonic pH, and are involved in gluconeogenesis and cholesterol synthesis [43].

## 4. Neuronal Factors of GI Disorders in ASD

### 4.1. The Enteric Nervous System (ENS)

The autonomic nervous system (ANS) includes the parasympathetic and sympathetic nervous systems, as well as the ENS (Figure 2) [44]. The ENS is the largest and most complex part of the peripheral (PNS) nervous system, as well as the ANS, in vertebrates [45]. The ENS is synchronized with the CNS, the other branches of the ANS, and the gastro-entero-pancreatic system to control the GI tract. The ENS can generate reflexive gut contractile activity independently of the rest of the GI tract’s innervation. This unique capability differentiates the ENS from other parts of the PNS [45]. The ENS is the primary initiator of complex motility patterns and is often identified as an additional brain [46].

Humans have approximately 400–600 million enteric neurons. This number roughly equivalent to the number of neurons in the spinal cord and exceeds the combined total of all the sympathetic and parasympathetic ganglia [47]. The ENS primarily originates from the vagal neural tube, with additional contributions from the sacral and upper thoracic regions, all of which contain neural crest cells [48]. As a part of the PNS, it comprises a complex network of neurons with few glial cells. In mammals, the ENS includes two ganglionated plexuses: the submucosal (Meissner’s) plexus, situated between the muscular mucosa and circular muscle, and the myenteric (Auerbach’s) plexus, located between the circular and longitudinal muscle (Figure 2). The submucosal plexus neurons primarily regulate the secretion of mucus, enzymes, bicarbonate, water, and electrolytes. They also regulate vasodilation and absorption. The myenteric plexus neurons are primarily involved in regulating intestinal contractile patterns and in synchronizing these patterns with other intestinal behaviors [2]. Enteric neurons from both plexuses comprise a diverse population of neurons, including sensory neurons, ascending and descending interneurons, and motor neurons [2]. The mucosal endings of sensory neurons are separated from the luminal contents by a continuous epithelial lining. When luminal contents bind to receptors on enteroendocrine cells, these cells release messenger molecules from their basolateral surfaces, activating enteric, vagal, and spinal sensory neurons [49]. For example, neural and endocrine integration is essential in gastric acid secretion. Intrinsic sensory enteric neurons detect signals from the intestinal wall and transmit the information to both ascending and descending enteric interneurons. The ascending interneurons typically synapse with excitatory motor neurons, while the descending interneurons connect with inhibitory motor neurons. Through these pathways, the ENS coordinates and regulates appropriate physiological responses in the gut. The ENS regulates the complex and essential functions of the GI tract, including breaking down food into absorbable nutrients and absorbing them, eliminating waste, and defending the GI tract against toxins, physical damage, and irritants. The regulation is performed by determining the movement patterns of the GI tract, controlling the secretion of gastric acid, regulating the movement of fluid across the epithelium, modifying nutrient processing, and signaling the immune and endocrine systems of the gut [47]. The extent to which the ENS is required for coordinated muscle function depends on the region of the GI tract and the physiological conditions.

The CNS largely determines esophageal peristalsis through the vagal motor neurons, although these neurons also form synapses with enteric neurons [47,50]. When food reaches the distal part of the esophageal body, the lower esophageal sphincter relaxes through a descending inhibitory reflex primarily mediated by the vagus nerve. The final motor neurons in this reflex are enteric neurons, predominantly nitrergic [47]. Much of the neural control of the stomach is dependent on vago-vagal reflexes (like the esophagus), and the caudal rhythmic contractile waves of the stomach are generated in the muscle through the interstitial cells of Cajal [48]—but neural control of the ENS in the stomach has not been demonstrated yet. By contrast, the ENS has predominant control of the motility of the small and large intestines. The brain and spinal cord direct the ENS activity of the intestine only in the colorectal region. Control centers in the CNS regulate the propulsive reflexes of the distal colon and the rectum, and defecation is triggered through the defecation center in the lumbosacral spinal cord [51].

### 4.2. Sympathetic Nervous System Overactivation and Dysbiosis That Is Unrelated to Nutritional Habits

ASD is frequently associated with GI dysbiosis unrelated to nutritional habits [52,53]. Parasympathetic (cholinergic) stimulation increases the activity of the ENS, whereas sympathetic (noradrenaline) stimulation inhibits it. The sympathetic branch of the ANS is overactivated in ASD, primarily due to a deficit in parasympathetic activity [54,55,56,57]. It creates an autonomic imbalance and disturbs the regulation of the gut–brain axis. Paneth cells are located throughout the intestinal villi and are essential in producing, storing, and secreting various antimicrobial peptides [58,59,60]. Cholinergic release from the parasympathetic nervous system causes the secretion of antimicrobial-rich granules, and the attenuation of this stimulus weakens the host’s ability to enforce the mucosal barrier and retain commensals within the gut lumen [59]. This leads to chronic metabolic stress and increased production of reactive oxygen species, which provide oxygen to intestinal bacteria. As a result, the balance of the microbiome shifts in favor of facultative anaerobic and aerobic bacteria, at the expense of anaerobic and facultative aerobic species. This shift helps explain the medium-term effectiveness of microbiota transfer therapy in treating GI symptoms associated with ASD [57].

### 4.3. iPSC-Based Approaches to Study ASD and ENS Dysregulation

Investigating the autistic ENS at the cellular and molecular levels is crucial for understanding how ASD affects the structure and function of the human ENS, as well as how the ENS contributes to ASD pathogenesis. However, obtaining enteric neurons or glial cells from living individuals with autism is not feasible. Induced pluripotent stem cells (iPSCs) are reprogrammed somatic cells capable of self-renewal and differentiation into any cell type from all three germ layers [61]. iPSC-derived GI organoids offer a promising alternative to rodent models and serve as a valuable tool for studying enteric neurons and glial cells genetically identical to those found in individuals with ASD.

Researchers have differentiated iPSCs into neurons from the CNS to study ASD at the molecular, cellular, and neuronal network levels [62,63,64,65,66,67,68]. Monitoring neural progenitor cells from individuals with idiopathic ASD revealed ASD-associated changes, including temporal dysregulation of specific gene networks and morphological growth acceleration. Furthermore, direct iPSC-to-cortical-neuron conversion, which skips the neural progenitor cell stage—the critical period for the ASD signature to establish itself—prevented these changes from manifesting [66]. iPSC-derived cortical neurons from four different mutant lines—*SHANK3*, *GRIN2B*, *UBTF*, and chromosomal duplication in the 7q11.23 region—demonstrated hyperexcitability and early maturation compared to control neurons derived from family members without ASD in 3–5 weeks of differentiation. The hyperexcitability and early maturation were expressed in increased sodium currents, increased amplitude and rate of excitatory post-synaptic currents, and a greater number of evoked action potentials in response to current stimulation [68]. This proves that the cortex may have convergent pathological mechanisms of ASD despite the different mutations.

The findings regarding hyperexcitability were similar in iPSC-derived neurons from the forebrain of individuals homozygous for the gain-of-function *KCNT1* ASD-associated mutation P924L, which exhibited hyperexcitability in spontaneously active neuron networks, elevated hyperpolarization amplitude following the action potential, and a shortened action potential duration in these neurons [69]. Immature iPSC-derived dentate gyrus granule neurons hemizygous for the ASD-associated A350V *IQSEC2* mutation also exhibited hyperexcitability, characterized by increased sodium and potassium currents, as well as a significant reduction in the number of inhibitory neurons. Moreover, they exhibited dysregulation of genes involved in development and differentiation. However, as the neurons matured, they became hypoexcitable. The mature *IQSEC2*-mutant neurons exhibited lower sodium and potassium currents, as well as reduced synaptic and network activity frequencies. Additionally, the expression of genes related to synaptic transmission and neuronal development decreased. The mature mutant neurons were less viable and had reduced expression of surface AMPA receptors compared to the control neurons derived from CRISPR/Cas9-corrected iPSCs [70].

ASD-associated mutations alter not only neurons but also glial cells. iPSC-derived oligodendrocyte progenitor cells with InsG3680 *SHANK3* mutation demonstrated impaired glutamate signaling, including a lower calcium activity rate, and exhibited impaired myelination [65]. The findings about the CNS can suggest ideas about how to research autistic ENS using iPSC models.

Decreased zinc uptake transporters ZIP2 and ZIP4 expressions were revealed in enterocytes of individuals with Phelan-McDermid syndrome (PMS), which is caused by the 22q13 deletion and includes ASD symptoms [71], at the mRNA and protein levels [72]. In addition, Dana Leavitt et al. successfully differentiated iPSCs from individuals with PMS into GI organoids in 2019 [73]. However, no study has utilized such organoids or two-dimensional iPSC cultures to investigate the autistic ENS. Exploring the autistic ENS by this approach could open new avenues for advancing ASD research.

## 5. GI-Related Genetic Factors in ASD

### 5.1. Pathway and Motif Analysis of ASD-Associated Genes Expressed in the ENS

Genetic variations are a significant risk factor for ASD. However, most of the ASD cases are considered sporadic [74]. There are ASD-associated mutations in genes that are expressed in both the CNS and the ENS [75,76,77,78,79]. Studies have utilized genetic models with such mutations, as well as patient-derived cells, blood and urine analyses, and GI imaging. The mutations in genes that are expressed both in the CNS and the ENS suggest a potential link between GI dysfunction and the pathophysiology of ASD. We cross-referenced ASD-associated genes from the GWAS catalog with genes expressed in the ENS using GTEx data. This analysis resulted in the identification of 387 such genes (Appendix A). Then, we performed a pathway analysis of the ASD-associated genes expressed in the ENS using various bioinformatic databases. Gene Ontology (GO) and Protein Analysis Through Evolutionary Relationships (PANTHER) databases showed an enrichment of such genes in pathway and cell component networks with a false discovery rate (FDR) < 0.05 (The full results are presented in Appendix A). Cell component pathways that showed an enrichment of these genes included cell projection, cell projection part, neuron projection, dendrite, axon, synapse, synapse part, postsynaptic membrane, growth cone, integral to endoplasmic reticulum membrane, intrinsic to endoplasmic reticulum membrane, site of polarized growth, neuronal cell body, cell junction, and Golgi apparatus (Figure 3a,b). Biological pathways in which these genes were enriched included antigen processing and presentation, the presentation of peptide and polysaccharide antigens via MHC class II, the specification of anterior–posterior and dorsal-ventral axes, nervous system development, and the regulation of neurogenesis and axonogenesis (Figure 3c,d). Motif analysis of these genes revealed DNA sequence motifs, which serve as binding sites for transcription factors, proteins, and microRNA (Figure 3e). These motifs regulate various critical biological mechanisms, including the regulation of cell proliferation and differentiation (neurons, muscles, and hematopoietic cells), apoptosis, autophagy, insulin signaling, immunity, inflammation, allergic response, transporting of cations across cell membranes, epithelial-to-mesenchymal transition, oxidative stress signaling, and stress reactions (including hypothalamus–pituitary–adrenal axis), metabolism processes (like development and function of adipocytes), embryonic development, and even initiation of male sex determination.

### 5.2. Findings from Genetic Animal Models on GI and ENS Alterations in ASD

As mentioned above, animal genetic models with ASD-associated mutations in genes that are expressed in both the CNS and the ENS demonstrate alterations in the digestive system and the ENS. Studies confirm that many genes that affect the structure and function of the CNS also affect the ENS and may lead to digestive symptoms. In the following section, we will present examples of studies using genetic animal models that highlight ENS alterations with possible relevance to humans with ASD. The human methyl-CpG genes [80]. Mutations in methy;-CpG binding protein 2 (*MECP2*) cause Rett syndrome [81], which is characterized, among other symptoms, by autistic features [82]. The human *MECP2* gene encodes an epigenetic factor that binds to methylated DNA and regulates the expression of its target genes. In *MECP2*-null zebrafish larvae, more neutrophil infiltration into the digestive system was found compared to wild-type zebrafish larvae [83], meaning a higher level of GI inflammation.

**Figure 3 ijms-26-09580-f003:**
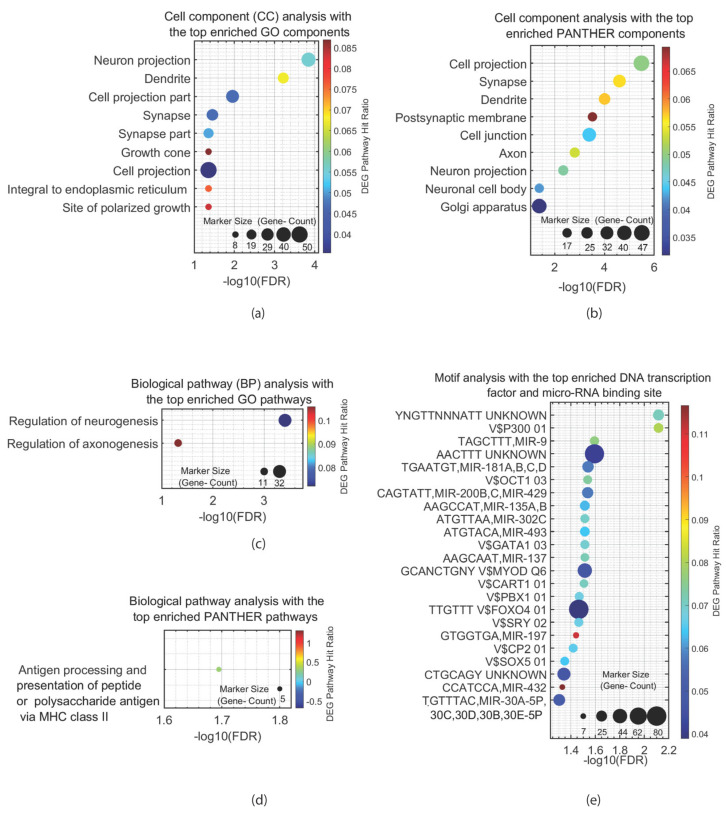
Pathway Analyses of ASD-Associated Genes Expressed in the ENS. Each panel shows the enriched pathways/cellular components/micro-RNAs in axis Y, and the −log of the false discovery rates (FDR) of the pathways/cellular/components/micro-RNAs in axis X. (**a**). Cell component analysis with the top-enriched Gene Ontology (GO) components for autism spectrum disorder (ASD)-associated genes expressed in the enteric nervous system (ENS) shows the highest enrichment for neuron projections and dendrites. (**b**). Cell component analysis with the top-enriched Protein Analysis Through Evolutionary Relationships (PANTHER) components for ASD-associated genes expressed in the ENS shows the highest enrichments for cell projections and synapses. (**c**). Biological pathway analysis with the top-enriched GO pathways for ASD-associated genes expressed in the ENS shows the highest enrichment for regulation of neurogenesis and axonogenesis. (**d**). Biological pathway analysis with the top-enriched PANTHER pathways for ASD-associated genes expressed in the ENS shows the highest enrichment for antigen processing and presentation of peptide or polysaccharide antigen via MHC class II. (**e**). Motif analysis with the top-enriched DNA transcription factor and micro-RNA binding site for ASD-associated genes expressed in the ENS shows the highest enrichment for an unknown motif and a binding site of the transcription factor *p300*, which is involved in the development of the brain, synapse, and memory [84].

Forkhead box protein 1 (*FOXP1*) is a transcription factor. Heterozygous individuals with a *FOXP1* deletion or loss-of-function have *FOXP1* syndrome, which often presents with autistic features. Mice with mutations in *FOXP1* consume less food and water and weigh less than wild-type mice. Rbms3, Nexn, and Wls, the target proteins of *FOXP1*, which were identified in the brain, are decreased in the mature esophagus of the mouse model. Furthermore, this mutation disrupts the relaxation of the lower esophageal sphincter, a process regulated by nitric oxide, leading to esophageal achalasia and impaired colonic contractions. These dysfunctions significantly prolong GI transit. Additionally, reduced muscle cell proliferation in both the esophagus and colon results in muscle atrophy [85].

Semaphorins are signaling proteins that play a key role in directing the growth and navigation of axons during neural development. They are essential for the CNS’s development, maturation, and function [86]. Semaphorine 5A (*SEMA5A*) mutations elevate the likelihood of ASD, according to research in humans and mouse models [87,88]. In rats, neurons of the ENS in the distal colon are positive for *SEMA5A* and its targets, PLEXIN A1 and A2. According to research on embryonic rat gut-derived ENS culture, the ASD-related missense *SEMA5A* mutation S956G impairs the functions of *SEMA5A*. The wild-type *SEMA5A* causes the axons of enteric neurons to be more complex and increases synaptic density compared to those of wild-type enteric neurons [89]. SYNAPSIN-1 phosphorylation at Ser9, Ser62, and Ser67 dissociates SYNAPSIN-1 from the actin filaments; Further, SYNAPSIN-1 phosphorylation at Ser603 dissociates SYNAPSIN-1 from actin filaments and synaptic vesicles [90]. Therefore, these phosphorylation modifications are essential for synaptic vesicle release in the active zone. *SEMA5A* causes the phosphorylation of SYNAPSIN-1 at Ser603, but not at Ser9, Ser62, and Ser67. As a result, the vesicle release decreases. Patch clamp experiments revealed a decrease in the number of neurons that wire spontaneously, the firing frequency, the amplitude of the action potential, and the postsynaptic miniature currents in neurons carrying the wild-type *SEMA5A*.

As noted, the mutation S956G impairs these characteristics. These results, along with the impairments in *SEMA5A*, show that *SEMA5A* is essential for regulating the connectivity of enteric neurons [89].

ENS progenitor cells express dual-specificity tyrosine phosphorylation-regulated kinase 1A (*DYRK1A*) during migration. Mature enteric neurons also express this gene, and particularly, it is highly expressed in primary cilia. Human loss-of-function mutations in *DYRK1A* that impair its function lead to ASD-like symptoms [91]. A mutation of *DYRK1A* in the diploid frog model *Xenopus tropicalis* that was generated during embryogenesis reduced the migration area of ENS progenitor cells and the expression of *SOX10*, which characterizes the early stage of the neural crest, compared to the wild-type organism. After the neural crest generation was completed and the neurons began to migrate, the inhibition of *DYRK1A* perturbed migration just as the mutation did in embryogenesis. These results indicate that *DYRK1A* is essential both for neural crest development and neuronal migration. The inhibition of *DYRK1A* in mature tadpoles led to a decrease in their defecation, indicating a decline in their gut motility [92]. Treatment with a serotonin selective reuptake inhibitor (SSRI) or an agonist for serotonin receptor 6 (5-HTR6) improved the gut motility in tadpoles whose *DYRK1* activity was inhibited, so that it was similar to that of control tadpoles [92].

Reducing the expression of the chromodomain-helicase-DNA-binding protein 2 and 8 (*CHD2* and *CHD8*, respectively), ASD-related chromatin remodelers, also significantly reduced the expression of *SOX10* in the same study using the Xenopus Tropicalis model [91]. The reduction in another ASD-related gene, *SYNGAP1*, which is involved in synaptic regulation of excitatory neurons and neuronal development [93], moderately reduced *SOX10* expression in the same model [91]. All these ASD-related mutations investigated in this article [92]—*CHD2*, *CHD8*, *DYRK1A*, and *SYNGAP1*—exhibited a convergent phenotype of reduced migration area of ENS progenitor cells, even if they did not mitigate *SOX10* expression.

Mutations in *SHANK3*, such as the 22q13 deletion that causes PMS, are related to ASD symptoms [71]. *SHANK3* is a key component of excitatory synapses, anchoring postsynaptic membrane receptors to the cytoskeleton. This connection is essential for the structural organization of the synapse and its function in neural transmission. *SHANK3* also binds additional ASD-related proteins, such as ADNP [94]. Mice that are homozygous for the *SHANK3B* knockout (KO) have a significantly different morphology of the intestinal epithelium, and their GI tract is more permeable than that of the wild-type mice. Additionally, the myenteric plexus is denser, with a higher number of mature neurons and neurons in a late stage of differentiation, compared to the wild-type mice. The contractions in the colon of these mice propagate to shorter distances and propagate more slowly than those of wild-type mice. Consequently, mice that are homozygous for *SHANK3* KO exhibit slower GI transit of food. Mice that are heterozygous for the *SHANK3B* KO exhibit these changes more moderately [95]. Similar findings regarding the transition time and contractions were observed in zebrafish larvae heterozygous or homozygous for *SHANK3A* or *SHANK3B* KO. Adult zebrafish that are heterozygous or homozygous for either *SHANK3A* or *SHANK3B* KO also exhibited more goblet cells in the gut tissue compared to wild-type adult zebrafish. It is interesting to note that zebrafish larvae that are heterozygous or homozygous for either *SHANK3A* or *SHANK3B* KOs exhibited fewer enteroendocrine cells that express serotonin than wild-type zebrafish larvae [96]. Along with the results of *DYRK1A* about treatment through the serotonergic system, the results indicate the high importance of serotonin in the development of ASD and digestive disorders.

Another review article summarizes a few ASD-related genes that are expressed in both the CNS and the ENS [2], including *CHD8* and *SHANK3*, and more examples. It briefly describes how each of the genes affects symptoms in humans and describes findings in animal models. One example is *CASPR2* (also called *CNTNAP2*), a type of neurexin that associates between cells in the synapse. *CASPR2* is expressed, among other tissues, in the sensory enteric neurons of mature mice. The colonic repetitive contractions in mice that are homozygous for *CASPR2* KO are 31% shorter than those of wild-type mice, and the colonic transit is accelerated [97]. *Neuroligin-3* (*NLGN3*) is another cell adhesion protein in the postsynapse, which supports synaptic maturation and transmission by interacting with both postsynaptic and intracellular proteins. In both mice with the knock-in of the R451C mutation in *NLGN3* and mice with *NLGN3* KO, the motility of the small intestine was faster than that of wild-type mice, and the GABA_A_ receptor of the colonic enteric neurons was more responsive. The small intestine of the knock-in mice also included more myenteric neurons. The colonic motility of the KO mice was faster than that of wild-type mice, and the diameter of their colon was longer [98,99]. The article in [2] presents an additional example of an ASD-related gene linked to the serotonergic system. *SLC6A4* encodes a serotonin reuptake transporter. The proliferation of crypt epithelial cells of mice with a knock-in of the ASD-related mutation Gly56Ala is increased compared to wild-type mice. Additionally, the knock-in mice exhibit ENS hypoplasia; their peristaltic reflex is less active, resulting in the GI tract transiting food more slowly than in wild-type mice. The phenotypes of gut motility can be contrary in different ASD-related genes. This suggests that different genes can lead to convergent outcomes (such as ASD development) with varying expressions (such as opposing GI symptoms).

Various techniques, such as selective breeding and clustered regularly interspaced short palindromic repeats (CRISPR), are used to generate animal genetic models [100,101,102]. Each model presents ASD differently, exhibiting varying behavioral phenotypes and comorbidities within the same species and across different species (such as faster versus slower food transit in the GI tract compared to control animals, also seen in the studies reported here). Moreover, even when the biological mechanism of the mutation is similar, some phenotypes may vary, also in humans. For example, in a comparison between behavioral phenotypes in 86 individuals with variants in the ASD-related gene *GRIN2B*, the phenotypes were diverse [103]. The variability in the phenotypes of rodent models may limit the validity of genetic animal models. Additionally, we must consider the design of behavioral paradigms, as well as the significant differences between humans and rodents. Still, findings from animal models may help find convergent mechanisms for efficient treatment development.

## 6. Metabolic and Dietary Factors That Influence GI Disorders in ASD

### 6.1. The Nutrition of Children with ASD and Its Influence on Gut Microbiota and GI Symptoms

A core symptom of ASD is resistance to change, and it is often manifested as emotional outbursts in response to even minor alterations in the environment or routine [104,105]. Indeed, according to the reports, 58–67% of parents of children with ASD and 22.9–69.1% of caregivers of children with ASD report that children with ASD are more likely to have food neophobia and/or selectivity. In contrast, only about 8.4–57.89% of parents and 1–37.1% of caregivers of children without ASD report food selectivity and/or neophobia of their children [105,106,107,108]. Food neophobia and food selectivity are normal in infants at the beginning of exposure to foods other than breast milk and its supplements. They are usually transient in children without ASD. However, in children with ASD, they often continue expressing longer and even forever [109]. In addition, about 90% of children with ASD exhibit atypical sensory responses (compared to about 33% of children without ASD) [110,111] due to hyper- or hyposensitivity to different stimuli caused by altered generation and processing of information in their somatosensory system [112,113,114]. A systematic review reports a positive correlation between food selectivity and impaired sensory processing [115]. Children with ASD often tend to prefer a single type/brand of food with specific textures, temperatures, smells, colors, and flavors [110]. A study of 279 patients with ASD and severe food selectivity reports that at least 67% of these children do not eat vegetables, and 27% of these children do not eat fruits. This study found a preference for ultra-processed, calorie-dense foods. More than 50% of children with ASD have significant nutritional deficiencies, especially in calcium, fiber, and vitamins D and E. Additionally, children with ASD who exhibit severe food selectivity often have an insufficient intake of protein, dietary fiber, and essential fatty acids, all of which can alter GI physiology and disrupt the balance of intestinal microbiota [106]. Therefore, food selectivity in ASD can lead to a significant nutritional imbalance that further contributes to GI problems [116,117,118].

### 6.2. Altered Metabolome in ASD

Zinc plays a vital role in the development and maintenance of the GI system, as well as in supporting synaptic plasticity within the ENS [119]. Zinc deficiency is significantly more common in individuals with ASD compared to peers without the condition [120].

Vitamin A is essential for the development of the CNS and PNS. Retinoic acid, the active metabolite of vitamin A, activates retinoic acid and/or retinoic X receptors in the nucleus, which bind to retinoic acid-responsive elements in the promoter regions of target genes and modulate their expression [121]. Children with ASD have lower retinal dehydrogenase 1 (an enzyme that creates retinoic acid) and lower retinoic acid levels in their serum compared to controls [122]. Moreover, vitamin A deficiency exacerbates core symptoms of ASD and is associated with GI problems (especially constipation) in children with ASD [123]. In a VPA-induced model of ASD, rats with gestational vitamin A deficiency exhibited more severe autistic-like behavior, longer GI transit time, and ENS dysplasia compared to ASD rats with normal vitamin A levels [124].

Many children with ASD have deficient activity of at least one disaccharidase (lactase, sucrase, maltase, palatinase, and glycolmylase [125]). Moreover, children with ASD and digestive disorders have a lower level of transcripts that encode disaccharidases and hexose transporters in intestinal biopsies compared to children with digestive disorders without ASD. This indicates that the enterocytes’ primary pathway to transport and digestion of carbohydrates is impaired. These results were associated with low levels of caudal-type homeobox 2 (CDX2) mRNA. CDX2 is a transcription factor that regulates the expression of sucrase isomaltase, lactase, glucose transporter 2 (GLUT2), and sodium-dependent glucose transporter (SGLT1). The expression of CDX2 and disaccharides in patients’ intestinal biopsies depends on the bacterial community structure in the gut [126].

According to blood and urine analyses of children with ASD and controls, there are abnormalities of various amino-acid-associated pathways, which are associated with neurotransmitter imbalances in children with ASD compared to control children [127]. Intermediate products of several amino acids, such as arginine, phenylalanine, tyrosine, tryptophan, and methionine, are increased in the urine samples of children with ASD. Some intermediate products also showed alterations in the blood samples of children with ASD [127]. Furthermore, loss-of-function mutations in the branched-chain ketoacid dehydrogenase kinase (BCKDK) enzyme in families with ASD, epilepsy, and intellectual disability cause a lower plasma concentration of branched-chain amino acids. KO Mice for BCKDK exhibited tremors, epileptic seizures, and hindlimb clasping. Pathway analysis of genes expressed in the cortex of KO and control mice revealed dysregulated pathways, including the brain-expressed amino acid transporters network [128].

An essential example of a perturbed amino acid pathway is tryptophan, whose metabolism disruption was related to the increase in two markers of reactive oxygen species—7,8-dihydroneopterin and neopterin—observed in the urine of 40 children with ASD, compared to control children [127,129]. Additionally, in the urine samples of children with ASD, there were also higher levels of arginine (a precursor of nitric oxide) and acetylarginine, and lower levels of antioxidants (such as anserine and carnosine) [130,131,132,133,134,135]. These findings may indicate excessive oxidative stress, a part of the pathophysiology of ASD [136]. The blood samples of children with ASD also exhibited decreased levels of antioxidants, such as superoxide dismutase, glutathione peroxidase, and 4-hydroxyphenyllactate, which are produced by lactobacilli and bifidobacteria [137]. The levels of serotonin, a product of tryptophan, were reported to be higher in urine samples of children with ASD. In contrast, the level of melatonin, a product of serotonin, was lower in their blood samples [135]. Melatonin deficiency is associated with sleep–wake rhythm disturbance in individuals with ASD, and studies have observed a mutual association between ASD symptoms, sleep disturbance, and digestive issues [138,139].

Children with ASD have lower levels of polyunsaturated fatty acids and higher levels of sphingosine-1 phosphate (a product of sphingomyelin) in their blood. This may indicate an abnormal metabolism of polyunsaturated fatty acids, which is associated with disturbances in neuronal structural and functional integrity [140] and an abnormal sphingomyelin metabolism associated with the abnormal development of white matter [141]. Perturbations in the peroxidation of lipids elevate advanced glycation end products and dityrosine in the plasma of children with ASD and lead to proteotoxic stress [142]. Furthermore, the increased level of dityrosine results from increased dual oxidase (DUOX) activity. Because DUOX is part of the immune system of the mucosal tissues, the overactivation of the enzyme may indicate an attempt to cope with pathogens, often due to a decrease in colonic barrier permeability. A reduction in the colon permeability enables pathogens to enter the colon and disturb the microbiome balance [143].

## 7. Immunological Factors of GI Disorders in ASD

### 7.1. Animal Research Immunological Maternal Models of the Autistic ENS: Strengths and Limitations

Researchers have developed various maternal environmental animal models of autism that effectively replicate the role of the digestive system in the condition. One of the models is MIA, which involves distinct antigens that activate the immune system during the animal’s pregnancy. In events that change the maternal-fetal immune environment, such as infection during pregnancy, immune signaling molecules may disrupt levels of neural development and increase the risk of offspring neurological disorders [144,145]. The MIA model includes protocols that vary in the type of immunogen used, timing, mode of delivery during pregnancy, and dose. All these parameters can determine the nature and severity of the phenotypes in the offspring [146,147]. The nature and severity of the phenotypes are also affected by the mouse strain [148], the individual maternal responsiveness within a strain [149], and the sex of the offspring [147,150]. These differences in methodological approaches challenge the comparison between studies. The diversity may also determine why MIA has a significantly different effect on some animal model pregnancies compared to others. Similarly, in humans, maternal infections of any type do not always result in brain or gut disorders [151]. However, most MIA protocols evaluate the prenatal immune challenge without additional genetic [152,153,154] or postnatal risk factors such as nutrition and postnatal environmental infections [155,156,157,158].

VPA is commonly used as an antiepileptic medication [159] and as a mood stabilizer for treating mood disorders [160]. It modulates neurotransmission [159,160,161] and epigenetically regulates gene expression by inhibiting histone deacetylase [162]. Exposure to VPA during pregnancy has been linked to increased risk for ASD in children (among other risks, including neural tube defects, developmental delay, and cognitive impairments). Deficient maternal care resulting from immunological stress during pregnancy is associated with a sex-dependent enhancement of conditioned fear in the offspring [163,164,165,166,167]. Rodents exposed prenatally to VPA show ASD-like behaviors such as lack of social interactions and play, elevated repetitive behaviors, and anxiety [168,169,170,171,172,173,174,175,176,177,178]. The VPA-induced ASD model recapitulates many behavioral and molecular deficits of idiopathic ASD, and different etiological factors leading to ASD may trigger the inhibition of histone deacetylase. The molecular deficits observed in this model resemble those found in individuals with idiopathic ASD, such as the downregulation of the *AKT/mTOR* pathway. However, the model does not capture all the molecular alterations associated with idiopathic ASD [172,179]. Additionally, the timing of VPA exposure during pregnancy influences the behavioral phenotypes observed in both mice and rats [172].

### 7.2. The Contribution of Gut Microbiota to ASD-Associated GI Issues

The gut microbiota is one of the environmental factors that has evolved significant interest in science, partially in ASD research, and mounting evidence supports its role in maintaining the gut and general human health [180]. One of the techniques to investigate it is the MIA model, which leads to dysbiosis in the gut microbiome [181,182,183]. Pre-conception microbiota transplantation can improve neurodevelopmental abnormalities of this model by inhibiting interleukin 17A (*IL-17A*) signaling (which is essential for immune responses, including inflammation) during pregnancy [184]. It is well-known that individuals with ASD often show imbalances in the gut microbiome compared to the general population. Changes in microbial metabolites, such as SCFAs and ammonia, correlate with ASD and its severity [185,186]. Analysis of the fecal microbiota of children with ASD shows an overgrowth of pathogenic strains and imbalances in the ratio between Bacteroidetes (Gram-negative, non-spore-forming, and anaerobic) and Firmicutes (Gram-positive, spore-forming, and obligate/facultative aerobic bacilli) [185,187,188]. Moreover, in most cases of dysbiosis observed in the feces, oral cavity, or saliva of children with ASD, there is considerable heterogeneity in both the affected microbial populations and the nature of the alterations. Adults with ASD exhibited lower alpha diversity in their fecal microbiota composition and a higher abundance of three bacterial 16S ribosomal RNA gene amplicon sequence variants than adults without ASD [189].

Studies observed an increased intestinal permeability in individuals with ASD [190]. Indeed, 75% of duodenal biopsies from children with ASD showed reduced expression of barrier-forming tight junction components (*CLDN1*, *OCLN*, and *TRIC*) compared to controls [191]. In addition, 66% of the biopsies from children with ASD showed increased mRNA and protein levels of pore-forming claudins (claudin-2, -10, and -15) compared to controls [191]. The elevated permeability of the intestines and changes in the gut microbiota may contribute to GI issues in ASD. Moreover, transferring fecal supernatants from adult ASD model mice to naïve mice led to reduced colonic barrier permeability compared to transfers from wild-type mice. This effect is attributed to the ASD model’s lower expression of tight junction proteins such as JAM-A, cingulin, and ZO-2, as well as decreased levels of the pro-inflammatory cytokines *IL-1β* and *TNF-α* in the proximal colon [51].

Changes in the gut microbiome can also lead to alterations in the ENS. Although the cellular structure of the ENS is established at birth, its functional maturation is shaped by the postnatal gut microbiota to which the newborn is exposed [192]. Indeed, the transfer of fecal supernatants from adult humans with ASD to mice reduced the expression of glial and neuronal proteins compared to the transfer of fecal supernatants from adult humans without ASD. The mice that received the fecal supernatants from humans with ASD showed a significant reduction in the proteins involved in neuronal connectivity in the ENS, such as βIII-tubulin and synapsin [189]. This suggests that changes in gut microbes may play a role in remodeling the ENS of individuals with ASD.

### 7.3. Differences in GI Inflammation Between Children with ASD and Control Peers

Esophago-gastro-duodenoscopy in 36 children with ASD and 26 control children showed that the Paneth cells of ASD patients were frequently enlarged, and they exhibited an increased number of Paneth cells per crypt compared to the controls [125]. In another ileocolonoscopy study on 148 children with ASD and 30 control children, 90% of the ASD children were found with ileo-colonic lymphoid nodular hyperplasia (LNH) versus 30% of the control children. Moreover, 68% of the children with ASD and LNH had moderate to severe LNH versus 15% of the control children with LNH. No correlation was found in this study between the presence and severity of ileal LNH and diet or age at the colonoscopy [193]. Immunohistochemical staining of children with ileal LNH confirms these findings [194]. Studies also discuss the molecular differences in GI inflammation between children with ASD and children with IBD without ASD. The study above [194] also showed that lymphocytic colitis in children with ASD is less severe than classical IBD, but with significantly increased basement membrane thickness and mucosal γδ T cell density compared to children with IBD (without ASD) [194]. Another study found a significant increase in CD3+CD4+ intraepithelial lymphocytes and CD19+ lamina propria B cells in children with ASD compared to children with IBD without ASD, in the duodenum, ileum, and colon [195]. In addition, transcriptome profiling of GI mucosal biopsy tissue obtained during ileocolonoscopy revealed that the GI mucosal gene expression profiles of autistic children with GI problems and those with IBD (without ASD) exhibit distinctive features, despite a significant overlap [196].

## 8. Conclusions

Collectively, the studies we have reviewed emphasize the critical bidirectional interaction between the digestive system and the CNS, which is essential in the pathogenesis of ASD and digestive disorders. This interaction highlights the profound connection between these conditions, showing they cannot be considered entirely separate and simply comorbid. The pathogeneses of autism and GI disorders mutually influence and reinforce each other. Furthermore, there is a strong bidirectional correlation between the severity of digestive issues and the intensity of autistic traits. The GI problems in ASD arise from various interconnected factors, including genetic and nutritional influences, an imbalanced gut microbiota, alterations in the metabolome, and overactivation of the sympathetic nervous system. Therefore, to deepen our understanding of the mechanisms behind autism, improve treatment options for ASD, and enhance the quality of life for individuals with ASD, it is crucial to refine the research models used continually.

## Figures and Tables

**Figure 1 ijms-26-09580-f001:**
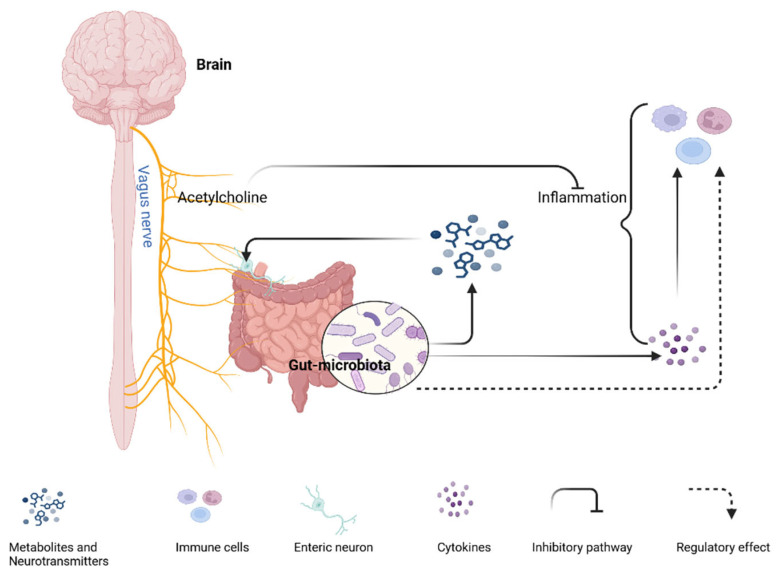
Scheme of Mechanisms of Interaction Between the Brain, Gut, and Immune System. The vagus nerve innervates the gut and attenuates systemic inflammation via acetylcholine. The gut microbiota produces neurotransmitters, metabolites, and cytokines that can influence neural circuits. The gut also creates serotonin. The gut microbiota regulates the differentiation and function of systemic immune cells, and the cytokines recruit the immune cells.

**Figure 2 ijms-26-09580-f002:**
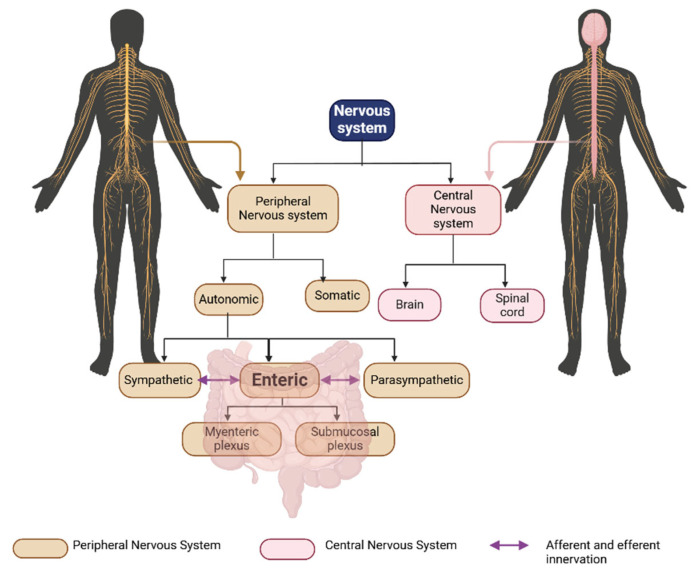
The enteric nervous system (ENS). A scheme shows the components of the ENS and its innervations. The nervous system has two components: the central and peripheral nervous systems (CNS and PNS, respectively). The PNS has three sub-components: the somatic, autonomic, and enteric nervous systems. The autonomic nervous system (ANS) is divided into the sympathetic and parasympathetic nervous system, and both components innervate the digestive system and contact the ENS with afferent and efferent neurons. The ENS is organized into two ganglionated plexuses: a submucosal and myenteric plexus.

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
