# Peer review of "Digestive Neurobiology in Autism: From Enteric and Central Nervous System Interactions to Shared Genetic Pathways"

_ijms, 2025, doi:10.3390/ijms26199580_

Round 1
Reviewer 1 Report
Comments and Suggestions for Authors
In this study, Robas et al summarized current knowledge about the pathogenetic mechanism of gastrointestinal (GI) symptoms in patients with autism spectrum disorder (ASD), with emphasis on the roles of gut-microbiota-brain axis and enteric nervous system (ENS).
This article readable, informative and interesting. The reviewer sees several issues to be addressed by the authors:
- This study seems to focus primarily on non-syndromic or “idiopathic” ASD (Line 445). This should be clearly stated in Abstract and/or Introduction, since the association of GI symptoms and ASD is much different between syndromic and non-syndromic ASD. In perinatal brain damage presenting with cerebral palsy, intellectual disability and developmental disorder, GI symptoms and ASD are simply comorbid (Line 633) in the majority of cases. In chromosomal abnormalities (e.g. Down, Angelman and Prader-Willi syndrome), GI symptoms and ASD are caused by the defect of multiple, adjacent genes. In single gene disorders with systemic involvement (e.g. mitochondrial disorders), both GI symptoms and ASD are directly caused by dysfunction of the same gene, being mutually independent. Therefore, the descriptions on the MECP2 (Rett syndrome), SHANK3 (Phelan-McDermid syndrome), PTEN (PTEN hamartoma tumor syndrome) and TSC1/2 (tuberous sclerosis complex) may not be very appropriate in this context.
- The “certain bacterial strains” is unclear. (Line 106)
- The contrast between TSC1/2 and PTEN (Lines 406-449) is not convincing because multiple previous studies have reported TSC rodent models showing deficits in social interaction.
- The “-“ between “without ASD” and “in the duodenum” seems unnecessary (Line 572), as well as that between “without ASD” and “in the duodenum” (Line 572).
- There are few “glial cells” in ENS because ENS belongs to the peripheral nervous system (Line 581).
Author Response
Dear reviewers,
First, we would like to thank the reviewers for their comments and feedback. They contributed to improving the article's quality and clarified its goal. We hope we can address all your concerns. The changes are highlighted in blue in the revised manuscript.
Reviewer 1:
Comment 1: "This study seems to focus primarily on non-syndromic or 'idiopathic' ASD (Line 445). This should be clearly stated in Abstract and/or Introduction…In single gene disorders with systemic involvement (e.g. mitochondrial disorders), both GI symptoms and ASD are directly caused by dysfunction of the same gene, being mutually independent. Therefore, the descriptions on the MECP2 (Rett syndrome), SHANK3 (Phelan-McDermid syndrome), PTEN (PTEN hamartoma tumor syndrome) and TSC1/2 (tuberous sclerosis complex) may not be very appropriate in this context." " We thank the reviewers for this critical comment. We added in the introduction that the knowledge in the article about gastrointestinal disorders in ASD is primarily related to non-syndromic/idiopathic ASD. Still, we would like to note that the genes we wrote about in the article (at least most of them) can also have variants in non-syndromic ASD, even if the variants are different from those found in syndromic ASD. We therefore left the section describing specific genes, while most of the article focuses on idiopathic ASD.
Comment 2: The “certain bacterial strains” is unclear. (Line 106) We appreciate the comment, and we added "in the presence of certain bacterial strains and inflammatory contexts”. We hope that the sentence is clearer now.
Comment 3: "The contrast between TSC1/2 and PTEN (Lines 406-449) is not convincing because multiple previous studies have reported TSC rodent models showing deficits in social interaction. Thank you for this comment. In response, we have now removed this example and instead discussed an ASD-related gene whose mutations can cause diverse phenotypes in individuals. We hope that this is a more suitable example.
Comment 4: "The “-“ between “without ASD” and “in the duodenum” seems unnecessary (Line 572), as well as that between “without ASD” and “in the duodenum” (Line 572)."Thank you for the suggestion. We replaced this "-" with a comma.
Comment 5: "There are few “glial cells” in ENS because ENS belongs to the peripheral nervous system (Line 581)".
Thank you for getting my attention. We added the word "few".

Reviewer 2 Report
Comments and Suggestions for Authors
Dear authors,
After reading your review on the topic "Digestive Dimensions of Autism: A Multiscale Exploration of Gut-Brain Interactions" - I have a few questions and comments:
- Line 29 - (1) – it is necessary to change the format of the references according to the journal’s rules; the source number should be written in square brackets.
- The aim of the work is missing. Please formulate the aim of your literature review.
- The novelty of this review is not entirely clear to me; at least, it is not stated in the Introduction section. As far as I know, there are several reviews related to the gut microbiome and autism spectrum disorders:
- Góralczyk-BiÅ„kowska A, Szmajda-Krygier D, KozÅ‚owska E. The Microbiota-Gut-Brain Axis in Psychiatric Disorders. Int J Mol Sci. 2022 Sep 24;23(19):11245. doi: 10.3390/ijms231911245. PMID: 36232548; PMCID: PMC9570195.
- Taniya MA, Chung HJ, Al Mamun A, Alam S, Aziz MA, Emon NU, Islam MM, Hong SS, Podder BR, Ara Mimi A, Aktar Suchi S, Xiao J. Role of Gut Microbiome in Autism Spectrum Disorder and Its Therapeutic Regulation. Front Cell Infect Microbiol. 2022 Jul 22;12:915701. doi: 10.3389/fcimb.2022.915701. PMID: 35937689; PMCID: PMC9355470.
- Alharthi A, Alhazmi S, Alburae N, Bahieldin A. The Human Gut Microbiome as a Potential Factor in Autism Spectrum Disorder. Int J Mol Sci. 2022 Jan 25;23(3):1363. doi: 10.3390/ijms23031363. PMID: 35163286; PMCID: PMC8835713.
- Zang Y, Lai X, Li C, Ding D, Wang Y, Zhu Y. The Role of Gut Microbiota in Various Neurological and Psychiatric Disorders-An Evidence Mapping Based on Quantified Evidence. Mediators Inflamm. 2023 Feb 8;2023:5127157. doi: 10.1155/2023/5127157. PMID: 36816743; PMCID: PMC9936509.
How is your review different from the others? If there is some uniqueness, it should be added to the Introduction section.
- Line 128 – “Spore-forming microbes…” – incorrect phrasing, better to replace with “Spore-forming bacteria” or “Spore-forming microorganisms.”
- Figure 1 – it is necessary to correct the caption of the figure, change the title of the figure, for example, “Scheme of mechanisms of interaction between the brain, gut, and immune system.” And the text in lines 140-143 “The vagus nerve innervates the gut and attenuates systemic inflammation via acetylcholine. The gut microbiota produces neurotransmitters, metabolites, and cytokines - that can influence neural circuits. The gut also creates serotonin. The gut microbiota regulates the differentiation and function of systemic immune cells, and the cytokines recruit the immune cells.” – it is better to move this text higher in the text, where the reference to figure 1 is given. Or it can remain as a description to the figure, additionally adding the figure title.
- Line 150 – “IL-17A” – it is necessary to explain what this is and why you mention it in the text.
- Figure 3 – the inscriptions are very small and difficult to read. In the legend of the figure, it is necessary to add explanations for all abbreviations. The axes on the graphs should be signed or clarified in the legend. General recommendation for figure 3: it is recommended to increase font sizes, make separate detailed explanations for each panel, unify the style of color coding, and add explanations of all terms and parameters used (at least in the caption under the figure).
- Lines 553-554 – “A reduction in the colon permeability enables pathogens to enter the colon and disturb the microbiome balance (184).” – I believe you made a mistake; perhaps an increase in permeability promotes the penetration of pathogens?
- I would recommend clarifying and correcting the title of the work. The current title “Digestive Dimensions of Autism: A Multiscale Exploration of Gut-Brain Interactions” suggests that the review focuses on brain-gut interaction and primarily on the connection between gut microbiome and central nervous system, which does not fully correspond to the content of your review.
In your work, not only interactions between brain and gut are considered, but also features of the gut nervous system, as well as genetic mutations associated both with autism and gastrointestinal disorders. However, neither in the Introduction section nor in the title is this explicitly reflected.
Since the connection between gut and brain is a bidirectional system including neural (mainly vagus nerve), humoral, immune, and metabolic signaling pathways, it is advisable to structure the manuscript considering these levels of interactions. Such separation will make the text clearer and help readers better understand the main idea of your study.
Author Response
Dear reviewers,
First, we would like to thank the reviewers for their comments and feedback. They contributed to improving the article's quality and clarified its goal. We hope we can address all your concerns. The changes are highlighted in blue in the revised manuscript.
Reviewer 2:
Comment 1: "Line 29 - (1) – it is necessary to change the format of the references according to the journal’s rules; the source number should be written in square brackets." Thanks for your comment. We changed to the journal’s format.
Comment 2: The aim of the work is missing. Please formulate the aim of your literature review."
Comment 3: The novelty of this review is not entirely clear to me; at least, it is not stated in the Introduction section. As far as I know, there are several reviews related to the gut microbiome and autism spectrum disorders: […] How is your review different from the others? If there is some uniqueness, it should be added to the Introduction section." We thank the reviewer for this critical comment. We added explanations at the end of the introduction to better clarify the review’s aims and uniqueness.
Comment 4: "Line 128 – “Spore-forming microbes…” – incorrect phrasing, better to replace with “Spore-forming bacteria” or “Spore-forming microorganisms."
Thanks for this comment. The word "microbes" was replaced by "microorganisms".
Comment 5: "Figure 1 – it is necessary to correct the caption of the figure, change the title of the figure, for example, “Scheme of mechanisms of interaction between the brain, gut, and immune system.” And the text in lines 140-143 “The vagus nerve innervates the gut and attenuates systemic inflammation via acetylcholine. The gut microbiota produces neurotransmitters, metabolites, and cytokines - that can influence neural circuits. The gut also creates serotonin. The gut microbiota regulates the differentiation and function of systemic immune cells, and the cytokines recruit the immune cells.” – it is better to move this text higher in the text, where the reference to figure 1 is given. Or it can remain as a description to the figure, additionally adding the figure title." We really appreciate these ideas. We have revised the title as suggested.
Comment 6 – "Line 150 – “IL-17A” – it is necessary to explain what this is and why you mention it in the text." Thank you for this suggestion. We added a sentence explaining what it is and its function.
Comment 7 – "Figure 3 – the inscriptions are very small and difficult to read. In the legend of the figure, it is necessary to add explanations for all abbreviations. The axes on the graphs should be signed or clarified in the legend. General recommendation for figure 3: it is recommended to increase font sizes, make separate detailed explanations for each panel, unify the style of color coding, and add explanations of all terms and parameters used (at least in the caption under the figure)." We thank the reviewer for pointing out this issue. We have increased the font size in Figure 3 to improve readability, and all abbreviations have been clearly explained in the figure legend. In all panels, the x-axis represents the FDR (False Discovery Rate) values, while the y-axis lists the names of pathways, and we have added this in the legend. Regarding the color coding, we would like to clarify that a uniform color scale across all subfigures is not feasible because the range of DEG (Differentially Expressed Genes) pathway hit counts for which the color coding is used, differs between panels. Similarly, the marker size scales vary across subfigures due to differences in the range of data values; therefore, we have provided separate size scales for each panel to ensure accurate interpretation.
Comment 8 – "Lines 553-554 – “A reduction in the colon permeability enables pathogens to enter the colon and disturb the microbiome balance (184).” – I believe you made a mistake; perhaps an increase in permeability promotes the penetration of pathogens?"
Thank you for this comment. The sentence and the reference were checked, and the mistake was corrected.
Comment 9 – "I would recommend clarifying and correcting the title of the work. The current title “Digestive Dimensions of Autism: A Multiscale Exploration of Gut-Brain Interactions” suggests that the review focuses on brain-gut interaction and primarily on the connection between gut microbiome and central nervous system, which does not fully correspond to the content of your review. In your work, not only interactions between brain and gut are considered, but also features of the gut nervous system, as well as genetic mutations associated both with autism and gastrointestinal disorders. However, neither in the Introduction section nor in the title is this explicitly reflected." We appreciated and accepted the suggestion. After revisions and clarifications in the introduction, we chose to change the title and call the article: Digestive Neurobiology in Autism: From Enteric and Central Nervous System Interactions to Shared Genetic Pathways. "Since the connection between gut and brain is a bidirectional system including neural (mainly vagus nerve), humoral, immune, and metabolic signaling pathways, it is advisable to structure the manuscript considering these levels of interactions." We took this comment seriously and appreciated it. Therefore, we decided to change the order of the chapters after the general explanation about the gut-brain-microbiota axis according to this division: . "Neuronal Factors of GI Disorders in ASD", "GI-related Genetic Factors in ASD", "Metabolic and Dietary Factors That Influence GI Disorders in ASD", and "Immunological Factors of GI Disorders in ASD". All these titles are marked in light blue, and all the chapters that we rearranged are marked in yellow. Finally, the conclusion of the whole article appears.
Regarding the additional comment we got: "After carefully rechecking it, we have noticed that references [108,110,145] cites papers that have been corrected. According to MDPI guidelines, retracted/corrected work cannot be used as evidence or scientific support for results." Thank you for this comment. We revised and corrected the references.

Round 2
Reviewer 2 Report
Comments and Suggestions for Authors
Dear Authors,
Thank you for the work done and the corrections made. Most of the comments have been addressed, and the manuscript has improved significantly.However, upon the final review, I identified several minor technical issues that require further correction. Please pay attention to them in the next revision.
- Please remove the bold font from some references on page 2.
- Line 156 – please make the heading bold.
- Lines 311-319 – change the bold font to regular.
- Lines 320-321 – please make the section heading bold, consistent with all other headings.
- Line 362 – “Pathway Analyses of ASD-Associated Genes Expressed in the ENS.” – please make the font non-bold, consistent with the captions of other figures.